# The Search of a Malaria Vaccine: The Time for Modified Immuno-Potentiating Probes

**DOI:** 10.3390/vaccines9020115

**Published:** 2021-02-02

**Authors:** José Manuel Lozano, Zully Rodríguez Parra, Salvador Hernández-Martínez, Maria Fernanda Yasnot-Acosta, Angela Patricia Rojas, Luz Stella Marín-Waldo, Juan Edilberto Rincón

**Affiliations:** 1Grupo de Investigación Mimetismo Molecular de los Agentes Infecciosos, Departamento de Farmacia, Universidad Nacional de Colombia—Sede Bogotá, 111321 Bogota, Colombia; zjrodriguezp@unal.edu.co; 2Dirección de Infección e Inmunidad, Centro de Investigaciones Sobre Enfermedades Infecciosas, Instituto Nacional de Salud Pública, 62508 Cuernavaca, Morelos, Mexico; shernand@insp.mx; 3Grupo de Investigaciones Microbiológicas y Biomédicas de Córdoba, Universidad de Córdoba, 230002 Monteria, Colombia; myasnot@correo.unicordoba.edu.co; 4Grupo de Investigación Biología Celular y Autoinmuniad, Departamento de Farmacia, Universidad Nacional de Colombia-Sede Bogotá, 111321 Bogota, Colombia; aprojasr@unal.edu.co; 5Centro de Salud San Pedro Claver, Nuqui, 276057 Choco, Colombia; lustmawa@hotmail.com; 6Departamento de Ingeniería y Mecatrónica, Universidad Nacional de Colombia-Sede Bogotá, 111321 Bogota, Colombia; jerinconp@unal.edu.co

**Keywords:** malaria vaccine, *Plasmodium* spp., structurally modified antigen, non-natural elements

## Abstract

Malaria is a deadly disease that takes the lives of more than 420,000 people a year and is responsible for more than 229 million clinical cases globally. In 2019, 95% of malaria morbidity occurred in African countries. The development of a highly protective vaccine is an urgent task that remains to be solved. Many vaccine candidates have been developed, from the use of the entire attenuated and irradiated pre-erythrocytic parasite forms (or recombinantly expressed antigens thereof) to synthetic candidates formulated in a variety of adjuvants and delivery systems, however these have unfortunately proven a limited efficacy. At present, some vaccine candidates are finishing safety and protective efficacy trials, such as the PfSPZ and the RTS,S/AS01 which are being introduced in Africa. We propose a strategy for introducing non-natural elements into target antigens representing key epitopes of *Plasmodium* spp. Accordingly, chemical strategies and knowledge of host immunity to *Plasmodium* spp. have served as the basis. Evidence is obtained after being tested in experimental rodent models for malaria infection and recognized for human sera from malaria-endemic regions. This encourages us to propose such an immune-potentiating strategy to be further considered in the search for new vaccine candidates.

## 1. Introduction

Malaria is a deadly and highly disabling disease accounted for 229 million clinical cases (2.8% by *P. vivax*) and 409 thousand deaths in 2019 [1]. This is a vector-borne disease to humans by the bites of Anopheline mosquitos infected with parasites of the *Plasmodium* genus. The incidence of malaria infection is increasing each year mainly due to accelerated climate change, provoking the mosquito adaptation to new altitudes and the increasing resistance of *Plasmodium* spp. to antimalarials.

Almost half of the world population is susceptible to infection by *Plasmodium* spp.— the causative agent of this disease—which suggests an economic and public health impact of incalculable proportions, as the disease’s impact on could lead to direct effects on welfare and education, as well as creating other indirect problems at the cultural, financial, and social levels [2,3].

To date, a large amount of literature from research groups around the world has been based on obtaining more potent and low toxicity drugs for the clinical management of *Plasmodium falciparum* malaria, mainly from natural sources, such as Artemisinin [4], as well as some others aimed at finding effective vaccines to prevent this deadly disease.

Vaccination is considered one of the most powerful means to improve the quality of life of human beings, with high impact and profitability in terms of public health [5]. Because of its use, a number of viral diseases have already been eradicated and vaccination in malaria has been regarded as a promising complementary public health measure [6]. However, effective vaccines against diseases caused by protozoa are not yet available for human use [7]. This seems to be directly related to the complex life cycles of these microorganisms, the different stages by which they suffer, and the large antigenic polymorphism, which would lead them to evade the immune response [8].

In the present work, we review the main vaccine candidates against malaria proposed to date, their characteristics, and some of their limitations, focused on new approaches that could contribute in the future to the development of a possible innovative candidate aimed at immunoprophylaxis. To fulfill our objective, information is collected with an emphasis on vaccine formulations that have become part of advanced clinical trials and have shown effects in humans, focusing especially on *P. falciparum*, but without ignoring the importance of also developing a vaccine for *P. vivax*—of which vaccine candidates are subjected to phase I clinical trials for [9,10] and whose particular capacity of hepatic-stages latency constitutes a challenging field of research [11]. Many decades of experience of a large number of worldwide research groups devoted to malaria vaccine development is represented by an important amount of vaccine candidates. Most have reached clinical trials, however in spite of their encouraging results, finding a promising vaccine is regarded as a high impact task, representing factual and conceptual challenges since the number of malaria deaths and clinical cases is worryingly growing. Considering such difficulties and malaria growing spreading, in this review, we are presenting some data generated in our research group regarding the development of *Plasmodium*-based modified antigens and their immunological profile as promising components of a synthetic vaccine candidate formulation. 

## 2. The Malaria Disease

### 2.1. Generalities

Malaria is a parasitic disease responsible for high levels of morbidity and mortality, distributed in the world neo-tropical regions including Africa, Asia, and the Americas. As can be seen in Figure 1A, malaria is present in around 91 countries registering 229 million cases and has cost the lives of about 409,000 people. Children under five years of age are the most vulnerable group affected by malaria; in 2019, they accounted for about two-thirds of all malaria deaths worldwide [1].

Malaria infection is produced by protozoa of the *Plasmodium* genus of which five species of parasites can infect humans, those being: *P. ovale, P. malariae, P. knowlesi, P. vivax*, and *P. falciparum*, the latter being the most pathogenic and deadly species. In a *Plasmodium* spp. infected individual, blocked blood vessels in the brain can occur, causing lethal effects [12]. *P. falciparum* is widely distributed in sub-Saharan Africa, representing 95% of the estimated cases of malaria worldwide in 2019.

Outside of Africa, *P. vivax* is the predominant malaria species in the Americas region accounting for 64% of malaria cases, 30% in South East Asia and 40% in the eastern Mediterranean regions [13]. Because of its complex biology, controlling infections by *P. vivax* is usually complicated given its capacity of hepatic latency, which can develop in weeks, months, or years after the initial infection. This state creates a possibility of producing new infections in blood with the respective clinical manifestations, and the possibility of transmission to another human host. Additionally, *P. vivax* possesses unique attributes to survive in various geographies associated with specific zoological and ecological characteristics, conditioning its global epidemiology, thus it is essential to control and eliminate infections caused by this species of *Plasmodium* [14,15].

The main differences between *Plasmodium* species are related to the microscopic morphological appearance they present, their potential for causing infection, the relapses they provoke, and their location in geographical areas. Clinical symptoms of malaria include acute fever episodes, headache, chills, and vomiting; symptoms that may appear between 7 and 10 days after human contact with the transmission vector. In many cases the infrastructure in rural endemic areas for reliable and rapid diagnosis is inadequate or nonexistent, yet a diagnosis made in time can avoid serious cases that can lead to death [16].

In spite of available antimalarial drug effectiveness, drug resistance has emerged, which can be attributed to factors such as the individual genetic immune profile, as well as pharmacokinetic and pharmacodynamic properties of the drug, failed monotherapy, among others [17]. Infections produced by resistant parasites lead to the spread of resistance to the next generation. In this way the parasitemia reappears, the anemic state of the patient worsens and the infection spreads, triggering therapeutic failures in later infections [18].

### 2.2. Plasmodium spp. Life Cycle

Malaria is transmitted to humans by the bite of the female *Anopheles* spp. mosquito that has previously contacted an infected human host. The transmission of the parasite between its two hosts involves significant morphological transformations that are constantly cyclically repeated, favored by the rainy season and influenced by conditions of temperature and humidity [19].

The cycle can begin with the pre-erythrocytic phase as observed in Figure 1B, which consists of the inoculation of sporozoites by the mosquito to a human host, which go through the dermis, pass through the blood circulation, and reach the hepatic parenchyma—this process takes between 5 and 30 min [20,21]. Unlike *P. falciparum, P. vivax* leads to the formation of quiescent liver forms called hypnozoites that remain in the host hepatocytes, with the ability to activate later. Then each sporozoite develops a schizont that releases between 10,000 and 30,000 merozoites into the bloodstream over a period of 2 to 10 days depending on the parasite species (Figure 1B) [22].

Following this, merozoites invade red blood cells after a series of interactions with membrane proteins of host cells, passing stages of interaction, reorientation, irreversible binding, and invasion, involving multiple antigens, this process is carried out in a few minutes [23,24]. 

*Plasmodium* spp. belongs to the apicomplexan phylum and its main characteristic is the presence of organelles such as rhoptries, micronemes, and dense granules, which are associated with invasion processes and are located at the apical end; proteins in these organelles allow parasite ligands to bind to certain receptors on the host cell membrane [25]. These interactions participate in the recognition and penetration of the parasite whose stages are committed to its’ survival and the ability to transmit the infection in organized patterns as reported [26]. Once inside erythrocytes, the parasite differentiates to an annular-shaped form as a young trophozoite, and later it reaches a configuration of mature trophozoite. Upon reaching maturity the merozoites are released from their host cell and infect new red blood cells during each erythrocytic cycle of the disease.

This phase is responsible for the clinical symptomatology of malaria and constitutes the asexual stage of the cycle called blood schizogony. In each red blood cell, schizogony is reduced and leads to the formation of 4 to 36 merozoites every 48–72 h, depending on the parasite species. The events carried out during the invasion process of the malaria parasite to its target cells appear to be very similar for all *Plasmodium* species [27].

Understanding this vital process increases the potential to block parasites through specific medications and vaccines at this point in their cycle, and it is estimated that in this way it will be possible to decrease the parasitic load and the complications of the disease. The erythrocytic phase is the most studied in the life cycle of *Plasmodium*, hence it is possible to maintain in vitro culture of these *Plasmodium* stages [28].

In the blood-stage, some merozoites differentiate into gametocytes (macro and microgametes) that are taken by a new vector to continue the sexual cycle of the parasite, favoring infection transmission into healthy individuals; during this process several morphological changes occur, as well as the expression of antigens.

### 2.3. Aspects on the Biology of the Malaria Transmission Vector (Anopheles spp.)

The *Plasmodium* sexual cycle begins with the bite of a female mosquito of an infected human host, from which it takes parasites as gametocyte forms, both male (microgametocytes) and female (macrogametocytes). Faced with this dramatic change (from a warm environment, free of other pathogens and rich in nutrients, to a hostile and colder one), the gametocytes must mature to initiate differentiation to gametes in a process called exflagellation. This process is favored mainly by conditions such as pH and temperature, in addition to the presence of a xanthurenic acid molecule which is synthesized in the mosquito´s parasite stages [29]. In the light of the invertebrate’s midgut, there is an environment conducive to these gametes in a transition stage that takes place in a period of 16 to 24 h (depending on the parasite species) constituting the diploid zygote that subsequently will differentiate into a mobile ookinete.

Under this form, to progress in the cycle, the parasite must overcome physical barriers such as the peritrophic matrix (PM, matrix that surrounds the food bolus with blood) and the midgut epithelium. To do this, the ookinete uses at least three types of locomotion and follows a combination of intercellular and intracellular routes [30].

Studies that analyzed in some detail the interactions between epithelial cells of *Anopheles stephensi* mosquitoes´ midgut with *P. berghei* ookinetes, established that these parasites, by invading columnar epithelial cells polarized with microvilli, cause them damage, favoring their apoptosis and rapid elimination (time bomb theory). At the same time, in a parallel and efficient way, the surrounding cells favor the repair of epithelial damage after an intense activity of actin polymerization. Invaded cells undergo alterations, among which are the substantial loss of microvilli, fragmentation of genomic DNA, and even the induction of the expression of nitric oxide synthase (NOS) [31].

It is estimated that from thousands of ingested gametocytes only 50 to 100 will become oocytes. This implies that throughout the *Plasmodium* spp. cycle, there is a loss of parasitic load as a consequence of the transit from one host to another and the migration of the parasite to different compartments within the mosquito. It is considered that the vector intestine represents one of the most hostile environments by which the parasite must go through to develop successfully, since in this organ, it undergoes, among others the action of digestive enzymes secreted by the intestinal epithelium [32]. The amount of ookinetes that colonize the intestinal light is a direct product from the number of fertilization events, since this is the only invasive stage that does not precede a replication event [33,34].

Morphologically, the ookinete is elongated and has an apical end in its structure which harbors protein-secreting organelles involved in motility and cell and tissue invasion, such as circumsporozoite surface protein (CSP), and thrombospondin-related adhesive protein (TRAP), among others. [35]. Recent studies have identified *Plasmodium* O-fucosyltransferase protein (POFUT2) which is responsible for O-glycosylations of CSP and TRAP; by interrupting this enzyme genetically, the ookinetes are attenuated to colonize the mosquito [36]. Similarly, it has been described that genetic expression attenuation of the gene encoding the PIMMS2 protein (*Plasmodium* invasion of mosquito midgut screen candidate 2)—a protein of *P. berghei* with orthologs in the genome of all *Plasmodium* spp. which is expressed specifically in zygotes and on the ookinetes’ surface—results in the loss of its function, despite the fact that it is possible to show normal development of the mobile ookinetes, and its ability to cross the midgut of an *An. gambiae* mosquito is reduced [37].

Once the ookinetes have crossed the epithelium, they are interspersed with the basal lamina of this organ in order to establish the infection and induce its transformation and maturation to sessile oocysts, which appear as protruding structures between muscle fibers of the midgut wall [38]. The oocyst is the only stage of the *Plasmodium* development that is extracellular in the parasite’s life cycle [35].

It has been shown that ookinete surface proteins including P25/28 and CTRP (Circumsporozoite- and TRAP-related protein) interact with the main components of the basal lamina such as collagen and laminin. It is believed that the latter plays a functional role in the development of the parasite from within the midgut of the mosquito, by employing RNAi techniques using a specific construct directed against the LANB2gene (laminin γ1) of *An. gambiae* a substantial reduction on the number of developed oocysts was successfully observed [39]. However, this is a matter of controversy due to results of in vitro tests in the absence of this protein in which oocysts are still developed [40].

Although oocysts are sessile forms, their evolution is dynamic and closely related to nutritional requirements provided by the invertebrate host. Oocysts in mature and early stages show notable differences in size, morphology, and composition of their membrane. [41].

As the parasite multiplies, the size of the oocyst increases dramatically, and the mature stages are substantially larger than the early stages [42].

Finally, the maturation of the oocysts, which occurs generally between 10 and 24 days (depending on the *Plasmodium* species), is a stage of growth and cell division known as sporogony, which leads to the production of thousands of sporozoites that are released into the hemolymph, where they are mobilized to invade the distal lobes of the salivary glands and finally be inoculated again into a vertebrate host to close the cycle.

Of the more than 400 species of Anopheles mosquitoes described, it is considered that approximately 17% are potential malaria vectors for humans, which means that only a limited number of species are competent vectors for humans [43]. In these insects some factors seem definitive, for example, longevity, pH of the midgut, temperature, and microbiota are critical for a successful infection [44]. In addition, genetic diversity, and the competence to develop the infection are determining factors, since it is known that not all females of a given species are equally susceptible to infection [45].

The mosquito immune response when invaded by parasites of *Plasmodium* genus has been studied to try to understand the complex molecular interactions between the host and the pathogen, which includes processes of recognition, signaling, signal transduction, and effector mechanisms [46,47].

The immune system of insects is efficient and is directed to the destruction of pathogens—their responses are generally similar to those of vertebrates [48]. To understand the immune response of *Anopheles* spp. against an infection, it is first necessary to know the cellular and humoral mechanisms that make it up. The cellular response is mediated mainly by cells of the hemolymph known as hemocytes, and the humoral side by antimicrobial molecules produced mainly by the fatty body. The hemocytes of different species of anophelines have been described based on their functional and molecular morphological and functional cytochemical characteristics, and even when there are some controversies regarding the number and types of cells that may occur, it is clear that they are the main effectors of phagocytosis processes, nodulations, and encapsulation of pathogens to invade the hemocoel. The humoral part involves different effector molecules, of which the most studied in mosquitoes are the phenol oxidase cascade (which culminates in the production of melanin necessary in the encapsulation processes), lysozymes, and antimicrobial peptides [49].

The study and understanding of the immune system characteristics of malaria vectors is essential for strategies focused on the prevention of the infection transmission. Understanding the factors involved in the infection of *Anopheles* spp. mosquitoes and the parasite’s sexual cycle development will make it possible to propose new antigens and strategies against malaria.

### 2.4. Immune Response in Malaria

The human immune system has mechanisms to defend against infectious diseases; this immunity depends on the joint and coordinated action of cells and molecules. When facing an infection with *Plasmodium*, the organism has defense mechanisms from both innate and adaptive immune responses. Innate immunity is known as the first line of defense, it is related to non-specific cellular and biochemical mechanisms, but adequate to face infections repeatedly, it includes a series of inflammatory pathways and the complement system, as well as the activation of natural killer cells (NK) mediated by cytokines [50]. In models of murine malaria, it has been demonstrated that innate immune responses (mediated by type I interferons (IFNs)and IFN-γ at the liver level) can limit the development of the parasite before its release into the bloodstream, therefore this could be considered as an interesting target in the development of pre-erythrocytic stage vaccines (Figure 2 and Appendix A) [51].

On the other hand, the adaptive immune response involves specificity and memory, that is, a response capable of distinguishing different substances that also involves more vigorous responses to repeated exposures to the same microbe. The only components of this type of response are lymphocytes and their secretion products which recognize anti-gens. In general terms, the immune response against the pre-erythrocytic stage is largely carried out by cells (NK) with the intervention of molecules such as nitric oxide, and cytokines such as IFN-γ, and IL-12. It can also lead to the CD8+ cytotoxic T lymphocytes (CTLs) stimulation through the interaction with helper T cells or alternatively activating CD4+ T helper lymphocytes (Th). This activation is given by the interaction of CD4 + T lymphocytes with certain subsets of Antigen Presenting Cells (APC) through the Major Histocompatibility Complex (MHC) class II, which can recognize and destroy infected target cells. [52], hence, these helper cells are required to induce antibody-dependent responses [53].

In the malaria infection blood phase, these antibodies are associated with reduced morbidity or protection against clinical symptoms in animal models, as epidemiological studies in adults has shown, where they are associated with controlled levels of parasitemia [54]. Antigens expressed in parasites or infected cells are recognized by different immunoglobulin isotypes of different subclasses (IgG1 and IgG3 in humans, IgG2a in mice). This can lead to a series of different effector mechanisms, for example, IgG1 and IgG3 mediate opsonization by phagocytosis, in addition, IgM is important for complement fixation, suggesting that antibodies can be associated with protection, through specific functional activities [55]. At this point, the secretion of some proinflammatory cytokines favors phagocytosis and the death of infected red blood cells by the action of macrophages. [56].

To ensure its survival, the malaria parasite has evolved mechanisms for escaping the immune system, such as antigenic variation, sequestration, and cytoadherence phenomena, and the diversity of sequences in non-repetitive regions, among others [57]. Some mechanisms of molecular secrecy occur during the transit of sporozoites from the place of the mosquito bite to the hepatocytes, during which the parasites molecularly protect vulnerable epitopes of their proteins, such as those known as “smoke screens”; mechanisms or others by which the parasite structurally resembles specific host antigens in order to avoid a given immune response. As a consequence, the parasite seems to efficiently modulate its antigenic protein, processing and releasing functional elements able to be used for successfully invading and infecting its target host-cells without being detected by the immune system [58]. Even today there are gaps in the understanding of how parasites evade the immune system, identifying these mechanisms will be crucial for the design of an effective vaccine against malaria [53].

The immune response against *Plasmodium* is species-specific and stage-specific, this means that acquired resistance against *P. falciparum* does not protect against infections by *P. vivax* and vice versa, as well as a protective immune response against sporozoite does not protect against infection of blood forms. Studies regarding malaria immunity—based on studies of animal models immunized with *Plasmodium*’s liver and blood stages—has led to evidence of T cell-mediated regulation and cytotoxic mechanisms (Appendix A). Evidence of protective T cell immunity against liver-stages after vaccination experiments with live sporozoites submitted to chloroquine treatment, as well as the role of CD4+ and CD8+ T lymphocytes activity of mice immunized with synthetic peptides has been published [59,60,61,62,63].

## 3. Classical Approaches for Malarial Vaccines

The development of vaccines against malaria has been considered a hard process since the identification of significant immunogenic proteins displaying the capacity of generating long-term antibody titers, knowing that the *Plasmodium* parasites have more than 5300 genes, grouped into 14 chromosomes increasing the conceptual complexity for a proper vaccine design [64]. Still, there is evidence that an immuno-prophylactic treatment for malaria could be possible due to three major matters: firstly, an apparent 90% protection of human volunteers by immunization with irradiated sporozoites [65,66], secondly, the passive transfer of immunoglobulins caused protection in adult volunteers [67] and thirdly, the acquired immunity specifically in populations from endemic areas [68,69] should be reinforced by immunoprophylaxis.

Efforts to develop malarial vaccines were initially based on employing attenuated or inactivated organisms as the vaccine, which often led to the reversion of virulence mainly in immune-compromised individuals. This fact has forced subsequent efforts towards more specific vaccines, such as those based on sub-units and purified artificial antigens from recombinant or synthetic origins. For these types of vaccines, it has been necessary to identify the pathogen proteomic composition, as well as the identity of the partial or complete antigens that could serve for inducing an eventual protective immunity, as well as search for appropriate formulations of highly immuno-stimulating adjuvants and delivery systems (Appendix A) [70].

With advances in molecular biology, approaches in the development of subunit vaccines of synthetic and recombinant origin boomed towards the 1980’s [71,72], nevertheless, these approaches have shown some disadvantages that later would give way to the exploration of increasingly innovative concepts to propose multicomponent, multistage, and multiantigen formulations that were potentially effective and viable [73]. By focusing on a single or a few antigens with an apparent functional role it is likely that this approach is too restrictive or broad, thus, multi-antigen vaccines require further investigation [74].

In general, the development of vaccines against malaria focuses on *P. falciparum*, as being the most lethal *Plasmodium* species, since of the more than 70 candidates proposed, about 30 have been tested in humans and only one of them passed to phase III clinical trials, while vaccine candidates for *P. vivax* malaria are much smaller in number, but not less important. (Figure 2, Appendix A). Given the co-endemicity of these two species in many regions, the eventual development of a multi-species vaccine regimen that can be used in a variety of high-risk populations would be the most appropriate, which could significantly reduce morbidity and mortality in endemic regions [75]. Currently, the arsenal of approaches to vaccines in malaria is diverse, in terms of the parasite life cycle stage target, and the technological platforms explored [76]. Among the main molecular antigen targets explored for vaccine development, some representative examples are summarized in Figure 1B, in line with some previous reports [77]. Each tested vaccine candidate consists of a unique combination of antigens, delivery platforms, and adjuvant systems consolidating critical information to encourage the development of new generations of vaccines against malaria [78]. A clear fact is the absence of a potent and fully protective vaccine against malaria, which is an urgent need that has to be promptly overcome since this lethal disease is getting worse, considering side factors such as accelerated climate change, as well as the pathogen and the transmission vectors, which are strengthening their resistance capacity to drugs and evolving more effective infection mechanisms.

### 3.1. Adjuvants and Vaccine Formulations

Due to the nature-made limited immunogenicity of some antigens, besides of favoring antigen presentation—as is the case of some sub-unit vaccine candidates—efforts towards the development of immuno-enhancers and novel adjuvant cocktail mixtures are aimed to reinforce the immune response against an antigen that, when co-administered with such an adjuvant is supposed to increase the vaccine potency, its immune quality, and duration of the immune response and perhaps memory T-cell stimulation, some of these adjuvants are reviewed in Appendix A.

Using adjuvants is aimed at (1) increasing the response to a vaccine, in the general population with special attention in populations with reduced response capacity due to age, disease, or therapeutic interventions, in relation to the increase in seroconversion rates, (2) reducing to smaller doses of the antigen, and (3) the number of natural contacts during immunization, would not have a direct influence on the adjuvants’ help function in the pathway from an innate immunity towards adaptive immune stability [79].

Among the conditions to have an ideal adjuvant are those related to costs, stability in its container for a long time, chemical stability, a defined action mechanism, and one of the most important, those that do not produce immediate or long-term adverse events [80]. Adjuvants can be classified regarding their action mechanism:Immuno-potentiators (IP): Lymphocyte stimulation depends on the antigen presentation in the regional lymph nodes by antigen presenting cells.Delivery systems (DS): Delivery systems for vaccines are combinations of an immunogen with some carrier-like compounds that lead to forming particulate systems thereof such as emulsions, microparticles, ISCOMs, and liposomes among others, and mainly function to target associated antigens into antigen-presenting cells (APC), including macrophages and dendritic cells. DS require the release of cytokines in soluble form or of membrane co-stimulatory molecules in antigen-presenting cells.Immuno-polarizing agents (IPz): Adjuvants that polarize the immune response in a direction required for protection, towards a given Th1, Th2, Th9, TfH, Th17, Th22, or Treg pattern, determined by the released cytokine profile among other factors [81].

In the treatment of human malaria, aluminum hydroxide and water-in-oil emulsions have been widely used, displaying a deposition effect that guarantees a slow and prolonged antigen release and has proven to recruit antigen-presenting cells at the inoculation site [80].

The family of oil adjuvants known as Montanide has demonstrated high potency in numerous experimental vaccine formulations tested in mice, rats, cats, dogs, and pigs, in combination with both synthetic peptides and recombinant viral antigens. Montanide ISA 720 has been studied to test human vaccine candidates against HIV and malaria [82]. Similar principles have been used for some veterinary vaccine applications, hence some adjuvant systems are based on adsorbed aluminum salts (aluminum hydroxide and aluminum phosphate salts) which have been used extensively against viral and bacterial agents, as well as in antiparasitic vaccines. Mineral oil emulsions have also been used as adjuvants with new generation Toll-like receptor agonists, formulated in liposomes, saponins, vitamin E, immune system complex stimulants (ISCOMs), virosomes, as well as different emulsions of oils of vegetable or animal origin [83].

The choice of an adjuvant can be decisive to obtain the best results for a given candidate vaccine, but is restricted to the available technical knowledge and supported by the concept that a correct combination between the antigen and the adjuvant could develop an effective vaccine considering some standard classical formulations [84]. Appendix A summarizes the most used adjuvants for vaccines.

Knowledge of mechanisms of innate immunity and its participation in the induction of specific immunity, particularly the description of the pattern recognition receptors (PRR), their ligands, and the molecular events that lead to the activation of intracellular signals constitute an important basis for designing vaccine adjuvants [85,86]. The above is becoming a relevant field of research in the face of the need to establish new safe and immuno-potentiating substances that can be licensed for use in humans. Some other relevant aspects regarding the role of adjuvant systems in vaccination were reviewed [87,88,89,90,91,92].

### 3.2. Plasmodium spp. Antigens Regarded as the Basis for a Vaccine Candidate

As has been discussed, the complexity of the *Plasmodium* spp. life cycle is a determinant for the development of a vaccine candidate, for that reason, several targets have been classically regarded (Figure 1B and Figure 2), three of which are Pre-erythrocyte, blood, and sexual (gametocytes) stages. These involve different approaches, some focus on infection prevention, others on the reduction of morbidity and mortality of asexual erythrocyte stages by decreasing multiplication, and others aim to prevent or block transmission to the mosquitos [7]. Also, the ability of *P. vivax* to maintain bloodstream infections even in the absence of active transmission makes it an attractive target for the development of vaccines for this species [11].

#### 3.2.1. Candidates Based on Pre-Erythrocyte Stages

The sporozoite form of *Plasmodium* plays a central role in the life cycle of the parasite, from the maturation of oocysts of the parasite in the transmission vector’s midgut, being a definitive process for the initial infection to the human host. The first approaches directed towards this stage of the life cycle of *Plasmodium* started from the use of the complete pathogen but attenuated by radiation (Appendix A). Ending the 1960s, it was demonstrated that vaccination using irradiated sporozoites was feasible, arguing that in tests with mice, a percentage of animals became protected against viable sporozoites [93]. In Macaca mulatta monkeys, partial protection was promoted with effect on the development of sporozoites delaying the pre-latent period [94], and in humans an effect of increasing the antibody titer against the main antigenic protein of the sporozoites was allowed [95]. Tests exhibited significant cytokine mediated responses, indicating that the protection induced by irradiated sporozoites is mainly based on the induction of CD8 + T cells, which were stimulated upon vaccination [96].

Consequently, the PfSPZ vaccine produced by Sanaria Inc, was developed which consists of metabolically active sporozoites, but not replicants, that is, sporozoites attenuated as the immunogen [96,97]. The results of the phase II clinical study for this vaccine candidate were recently published. In this trial, tolerability, safety, immunogenicity, and protective efficacy of direct administration by venous inoculation of three or five doses of vaccine in non-immune subjects was evaluated, demonstrating protection against heterologous and homologous malaria. These results provide the basis for the development of an optimized immunization regime to prevent malaria [98]. In addition to the previously mentioned approach, vaccines with genetically attenuated parasites (PfSPZ-GAP) with deletion in genes that regulate infective processes have been included and evaluated [99,100]. Other efforts show vaccine candidates in which the infectious agent is in vivo attenuated with the concomitant administration of antimalarial drugs such as chloroquine, which is called chemoprophylaxis or CVac vaccine [101,102].

Research groups from at least 15 countries are part of the consortium called I-PfSPZ-C (International PfSPZ Consortium) whose purpose is to advance the development of a vaccine that includes among its characteristics the ability to protect against natural transmission in the long term, with excellent safety and tolerability profile and that is also operationally viable for administration to the entire population. To establish a correlation between the immunological responses measured in these trials and to continue with operational matters for the development of phase III clinical trials, some tests are being conducted in countries such as the USA, Germany, Tanzania, Mali, Burkina Faso, Kenya, and Equatorial Guinea [103].

The CSP polypeptide has been regarded as the main surface antigen of *Plasmodium* spp. sporozoites. It has been proposed that it contributes to parasite development within the female mosquito as well as being active during infection [104]. Genes coding for CSP are located on *P. falciparum* chromosome 3. It is synthesized as a 50 to 70 kDa precursor which is subsequently processed into a mature surface protein of 40 to 60 kDa [58,105].

This antigen contains an immunodominant epitope being a repetitive Asn-Ala-Asn-Pro (NANP) amino-acid sequence in the protein’s central region—this has been regarded as highly immunogenic which occurs in different *Plasmodium* species. Despite exhibiting high polymorphism, different studies revealed that this region activates B lymphocytes producing antibodies that block culturing sporozoites, as well as cytotoxic lymphocytes.

#### 3.2.2. Plasmodium Blood Stages Antigens

Regardless of the complex parasite life cycle, malaria clinical manifestations only occur during the blood-stage and it has been long regarded that merozoite blockage at this point would prevent the invasion and development of the parasite within the red blood cell, providing protection to infected subjects. An efficient vaccine against these asexual stages should ideally promote a cellular response mediated by T lymphocytes with the capacity to inhibit parasite development into red blood cells, basically, it should promote a neutralizing antibody response acting as parasite entrance blockers. In some cases, cellular immune responses have been achieved against merozoite antigens, using viral vectors such as Chimpanzee Adenovirus 63 and modified attenuated viruses (ChAd63-MVA), but the lack of impact on the parasite growth rates on blood continues to drive interest in the induction of antibodies to generate a protective efficacy (Appendix A) [106].

It is important to bear in mind that in malaria-endemic regions, repeated exposure to *Plasmodium* infection can result in the development of naturally acquired immunity [107]. This is characterized by the development of strong antibody responses to parasites, particularly on blood stages. Many studies have attempted to establish a correlation between protective immunity against clinical manifestations and total levels of antibodies specific to merozoite antigens; these antibodies have broad specificities and functions, therefore, these antigen-antibody correlations remain largely unclear [55]. The ability of the *P. falciparum* merozoite to invade erythrocytes depends on specific interactions between the parasite and red blood cell receptors. Several proteins expressed on the merozoite’s surface which are actively involved in the invasive process have been identified. Among those, the 10 members of the Merozoite Surface Protein (MSP) family have been extensively studied. This group of molecules is not distributed homogeneously in the merozoite membrane, nor processed in a similar way, but are characterized by their structural diversity which includes domains involved in a variety of processes such as protein–protein interactions, essential for the initial formation of the merozoite as well as for contacting the red blood cells (RBCs) membrane (Figure 1B) [108].

The MSP-1 protein is one of the main *Plasmodium* surface antigens and one of the most abundant on the surface of merozoites. It is a 195 kDa protein, synthesized in schizogony, possessing conserved, semi-conserved, and variable sequences [109]. It is a precursor of protein fragments of 83, 30, 38, and 42 kDa product of a primary proteolytic degradation [110]. The 42 kDa C-terminal fragment undergoes a second cleavage processing to a 33 kDa soluble fragment and a second of 19 kDa which remains bound to the merozoite surface during invasion to RBCs [111]. The common hypothesis proposes that antibodies against MSP-1 could prevent the infection of red blood cells, which has been partially demonstrated in a study with an MSP-1 recombinant antigen tested in the *Aotus* monkey model, and its effectiveness was also dependent on the adequate selection of an adjuvant system in the vaccine formulation [112].

Furthermore, the MSP-2 antigen is an integral membrane protein of a molecular weight of approximately 45 kDa. It has a central region with significant variability between different *Plasmodium* isolates, and its *N*- and *C*-terminal ends are well conserved. Anti-MSP-2 antibodies are known to predominantly recognize the variable domains that can be associated with fewer episodes of fever and anemia, both symptoms known as indicators of morbidity. In a small efficacy study in Papua New Guinea, a vaccine incorporating the MSP-2 antigen reduced the density of parasites in participants [113], but considering the important polymorphism of this protein, a high degree of protection is not likely when used as the only component of a vaccine [114].

Moreover, the MSP-3 is a protein member of the MSP family of approximately 48 kDa and is a precursor of fragments of variable molecular weights. Its importance has been recognized, since antibodies directed against it could prevent invasion of merozoites [115]. The MSP-4 is a protein of about 40 kDa, located on the surface of the merozoite, with an *N*-terminal region composed of hydrophobic amino acids [116], it remains on the surface of the merozoites. during invasion and can be detected in the intraerythrocytic parasite [117]. The gene coding for MSP-4 is found on chromosome 2, located next to the gene that codes for the protein MSP-5 which is on the surface and apparently lacks antigenic diversity [118]. It is known that its stimulated antibodies are significantly associated with a lower incidence of clinical malaria [119]. The MSP-6 antigen is a precursor of a 36 kDa fragment that associates in complex with MSP-1. IgG antibodies from rabbits immunized with a recombinant MSP-6 have shown to inhibit the invasion of RBCs by *P. falciparum* strain 3D7 merozoites by 20% [120]. For its part, the MSP-7 protein is mainly hydrophilic consisting of 351 amino acids (41 kDa). The gene coding for the MSP-7 is transcribed in the asexual cycle, in mature blood stages, and accumulates at a time that coincides with MSP-1 expression [121]. Member MSP-8 is a protein whose predicted molecular mass is 69 kDa. It is expressed throughout the asexual life cycle of the parasite with the highest levels after 21 h in trophozoites and schizonts and is detected on the surface of the merozoite [122]. The MSP-9 component is located on the surface of the merozoite and in the parasitophorous vacuole of infected erythrocytes [123], it is also known as ABRA by its acronym Acid Basic Repeat Antigen. It is a highly conserved antigen in *P. falciparum*, which is recognized by antibodies. Synthetic peptides derived from its sequence and inoculated into rabbits have been shown to stimulate IgG antibodies which are capable of in vitro blocking merozoite invasion of human erythrocytes up to 90% [124]. As a recombinant protein ABRA has been recognized in sera from individuals from the malaria endemic area of India, which is of interest at the level of immune response [125]. The MSP-10 antigen is a 524 amino acid protein that is initially expressed as an 80 kDa protein and is further processed to a smaller 36 kDa form located at the apical end of merozoites. Like MSP-8, the protein sequence has asparagine repeating regions towards its *N*-terminus [126].

The proteins of the MSP family are closely related between *P. falciparum* orthologs, and studies with phylogenetic approaches present evidence of selection for their purification in the lineage that leads to *P. vivax,* therefore, these antigens that evolve under strong functional restrictions could become valuable candidates for vaccines [127]. 

The antigen known as SERA (Serine Repeat Antigen), constitutes a family of antigenic proteins of *Plasmodium*. Specifically, SERA-5 is one of this family’s antigens that has been used in human clinical trials in both adults and children [128] and is therefore important at the vaccine level. This protein is abundantly expressed in the *Plasmodium*’s parasitophorous vacuole as well as on the surface of the merozoite, for which its essential role in the life cycle of the parasite has been demonstrated [129].

GLURP (Glutamate rich protein) has a predicted molecular mass of 145 kDa and is also expressed on the surface of the merozoite, as well as in pre-erythrocytic stages. It has conserved, tandem repetitive regions that are capable of stimulating T and B lymphocytes [130]. It has been used to develop vaccines alone or in combination with other blood-stage antigens [131,132].

Interestingly the so-named VAR2CSA antigen is an important vaccine target to prevent placental malaria. VAR2CSA is the principal antigen ligand for chondroitin sulfate A in two allogeneic isolates of *P. falciparum* erythrocyte membrane protein 1 (PfEMP1), encoded by the var2csa gene has been regarded as the principal binding ligand associated with placental sequestration and binds to Chondroitin Sulfate A (CSA). As it has been recently reported, in malaria-endemic areas, pregnant women commonly suffer from placental malaria wherein *P. falciparum*-infected erythrocytes (IE) sequester in the placenta and often elicit an inflammatory infiltrate. PRIMVAC is a VAR2CSA-derived placental malaria vaccine candidate aiming to prevent serious clinical outcomes of *Plasmodium falciparum* infection during pregnancy. Thus, expectations regarding trial progress will be useful for malaria prevention [133].

#### 3.2.3. *Plasmodium* Antigens Expressed in Merozoite Rhoptries and Micronems

AMA-1 (Apical Membrane Antigen-1) is an integral membrane protein of approximately 80 kDa, conserved in the genus *Plasmodium*. It is expressed in both the merozoite and sporozoite stages. It is stored in micronemes and then translocated to the merozoite membrane, mediating the merozoite reorientation towards the RBCs and stimulating a macromolecular complex formation with a protein group located in the neck region of robe-shaped merozoites [26]. AMA-1 has also been related to the invasion of *Plasmodium* sporozoites to liver cells [134], this process involves secretion from micronemes of AMA-1 and the Thrombospondin-related adhesion protein TRAP.

The EBA-175 (erythrocyte-binding antigen) is a transmembrane protein also present in micronemes and characterized by revealing two conserved cysteine rich domains. It is expressed on the surface of the merozoite and involved in the invasion of the merozoite to RBCs by binding to its receptors via sialic acid. Assays with antibodies from rabbits inoculated with some of its fragments indicate invasion blocking activity [135]. Naturally acquired antibodies against EBA-175 have also been shown to inhibit the binding of its invasive ligands to their receptors on RBCs, and these inhibitory antibodies have been associated with a protective immunity [136]. Adults vaccinated with this antigen showed that it was well tolerated, safe, and immunogenic [137].

Studies in primates immunized with AMA-1 purified from *P. knowlesi*, evidenced some immunity with partial protection [138]. However, this fact does not appear to be reproducible for recombinant AMA-1 from *P. falciparum* [139,140].

Also, in the ring-shaped sites of other organelles of *P. falciparum* known as dense granules, the antigens RESA (ring-infected erythrocyte surface antigen), a 155 kDa protein and RIMA, (ring membrane antigen), among others have been characterized [141]. Studies related to their immunogenic properties are in progress. It is well known that these antigens are released after the invasion of merozoites in localized areas of the parasitophorous vacuolar space, before passing into the cytosol of newly invaded erythrocytes [142]. The RESA protein has been postulated to cross the membrane of the parasitophorous vacuole and interact with spectrin in the RBC cytoskeleton. Immunization assays with repetitive regions of this protein have induced antibodies that nevertheless decreased with time [143]. This protein has been used in multicomponent vaccines that also include MSP antigens.

In summary, the most representative blood phase antigens regarded as targets for human vaccine candidates that have reached clinical trials are MSP1, MSP2, MSP3, AMA1, EBA175, GLURP, PfEMP1 (VAR2CSA), and SERA. Generally, these have been presented as recombinantly expressed antigens and co-administered with different adjuvants and delivery systems as seen in Appendix A. Most of these candidates have been enrolled on phase I trials, a small number of candidate vaccines have reached phase II and large field trials to assess their safety and efficacy protection against naturally acquired malaria infections, while others have reached phase I / II trials assessing their protective efficacy against experimental infections in vaccinated volunteers [24,144]. Despite a large number of blood-stage vaccine candidates, almost all have been based on a limited number of antigens expressed in *Plasmodium* merozoites, and there is not yet a blood-stage vaccine that has reached a phase III clinical trial [145].

Finding a safe and highly effective malaria vaccine remains a challenge to be faced and solved especially considering the complexity of the *Plasmodium*’s life cycle, which seems to use quite elaborated evasion mechanisms to escape the host immunity, which probably includes its own antigens’ structure modulation which could be governed by so far unknown strategies and precisely employed by *Plasmodium* to avoid being targeted by a poorly stimulated immune response.

#### 3.2.4. Transmission-Blocking Vaccines

A third approach for the development of vaccines against malaria consists of transmission blockers, that is, they are directed to antigens from pre-fertilization stages (gametes) or antigens expressed once fertilization occurs, thus acting to prevent infection (Appendix A). In preclinical models, immunity induced by this type of vaccine—antibodies to antigens of the parasite in the sexual phase which are ingested by the mosquito after the bite—can inhibit the survival of the parasite in the midgut of the insect [146], which would allow generating a collective benefit in the long-term, which is why they are known as “altruistic vaccines”. Therefore, in this approach it is essential to identify antigenic proteins present in gametocytes, gametes, and ookinetes, among which Pfs25 stands out, a 25 kDa protein expressed on the surface of macrogametes, zygotes, and ookinetes, also a Pfs230 protein present in pre-fertilization periods and the Pfs48/45 gamete surface protein. The antigens’ names are derived from the parasite species which they come from i.e., Pfs, in this case, *P. falciparum*, and apparent molecular weight estimated by SDS-PAGE. Orthologous genes for these proteins have been found in other *Plasmodium* spp., including *P. vivax* (Pvs25), *P. berghei* (Pbs25), and *P. yoelii* (Pys25) [147]. Comparative studies of the blocking activity of these antigenic proteins support their use as vaccine candidates for the sexual stages of the parasite [148].

The Pfs25 is one of the best described and characterized transmission blocking antigenic proteins. A vaccine of recombinant origin of this antigen for *P. falciparum* administered with Montanide ISA 51 as adjuvant was evaluated in a controlled phase I study, finding levels of antibodies significantly different from the control in volunteers who completed the vaccination scheme, which could be related to the blockage of the infection, however systemic adverse events were also reported—including erythema nodosum—and were directly associated with the formulation, suggesting extended further studies [149]. To improve the demonstrated immunogenicity of Pfs25, a recombinant vaccine expressed in *Pichia pastoris* and conjugated with exoprotein A (EPA), called Pfs25M-EPA/Alhydrogel^®^, was subsequently proposed. The results of the phase I clinical trial carried out in the United States for this vaccine demonstrated the relationship between antibody titers and transmission blocking activity, however, the highest level of antibody titers was achieved at the highest dose after administration of the fourth immunization and decreased over time, considering that their functional activity was short-lived [150].

The Pfs230 is a 230 kDa protein belonging to a structural family characterized by partially conserved cysteine motifs. The gene coding for this protein was originally cloned from *P. falciparum* (strain 3D7) [151]. It is expressed on the surface of newly formed macrogametes and zygotes. It has been shown that antibodies against this protein block the transmission of the parasite to the mosquito vector [152]. This is an idea more recently reinforced in tests using rabbit antibodies versus a Pfs230 recombinant. This protein is regarded as having a central role in gamete–gamete interaction, so its evaluation in clinical trials would be a significant step [153].

Differential expression of the Pfs25 and Pfs230 antigens during the parasite’s sexual development may be useful to improve the efficacy of some vaccines when carrying out the co-administration of these antigens, significantly increasing the antibody response and therefore its functional activity. By its side, the Pfs48/45 participates in gamete–gamete recognition in blood-like tissues within the mosquito, this is expressed on the surface of gametocytes within the human host. The natural development of antibodies against this protein has been reported in individuals living in malaria endemic areas [154]. Vaccines targeting gametocytes and the parasite stages into the mosquito are relatively new, but may help in arresting transmission, bearing in mind that antibodies induced at this point are only a small subset of the targets of this approach. The use of existing datasets from transcriptomics and proteomics would emphasize the existing list of antigenic proteins to stimulate the transmission blockage and so open new pathways towards promising functional studies aimed to eradicate malaria [155]. 

Interestingly a *Plasmodium falciparum* antigen named Pfs47 is a strategic vaccine target. This is a member of the 6-cysteine family of proteins and is expressed on the surface of female gametocytes, gametes, zygotes, and ookinetes; hence it has been regarded as an important target for transmission-blocking vaccines. However, a recombinantly expressed protein rPfs47 did not show transmission-blocking activity (TBA), while its stimulated antibodies were effective tools of TBA. Therefore, the role of this *Plasmodium* target for eliciting functional antibodies became evident for developing malaria vaccine candidates [156].

#### 3.2.5. New Strategies for Immunopotentiation Antigenic Targets in Malaria

Those limitations and disadvantages already mentioned for the development of biological and recombinant vaccines against malaria, as well as the knowledge accumulated to date in this field, can be regarded as input information towards the development of new approaches for more potent vaccine candidates where synthetic epitope-antigens gain relevance. Peptide as vaccine candidates could be useful for immunization processes with single or multiple combinations of diverse molecules sizes and origins, designed with the purpose of reaching an optimal specificity to bind major histocompatibility complex (MHC) molecules, which once formed can then be recognized by T cell receptors and initiate a directed immune response by suppressing some natural requirements for antigen processing and presentation of protein epitopes [157].

Peptide-based vaccines represent a potential strategy for the prevention and treatment of pathogenic diseases, their low cost, synthetic feasibility, and inherent safety are all attractive features, however, they have remained largely unsuccessful due to the native peptides’ low immunogenicity, which comes mainly from the reduction of bioavailability due to proteolytic in vivo degradation [158,159].

Some genetically conserved proteins from different stages of the malaria parasite which could be immunologically relevant are usually silent to the host immune system and so escape their recognition.

Hence, to reveal its immunogenic nature, this non-recognizing phenotype must be overcome by incorporating modulating structure elements of non-natural origin. In consequence modified versions of such low immunogenic antigens now would potentiate their molecular recognition and functionality thus becoming useful for malaria prevention [160]. This promising approach is based on the development of modified peptide compounds called peptidomimetics (structurally modified peptides). Thus, it begins from a methodological design aimed at incorporating functional changes at the molecular level on the antigen back-bone as well as on its stereochemistry [161]. Therefore, some modified antigens have proven to be functional as avoiding or controlling the malaria infection as has been tested in experimental rodent and primate models [162]. Interestingly, despite being modified, this realm of novel molecules is designed to preserve the genetic information of the pathogen since amino-acid variations or mutations are not included. Some studies have even highlighted that a single replacement of a peptide-bond modulates the three-dimensional structure of the peptide-antigen and has proven to be sufficient to achieve greater protective capacity compared to the native antigen [161]. Obtaining such envisioned modified molecules based on antigenic regions of malaria has allowed establishing structural conformations that are evidenced to be associated with immunogenic properties, as already reported [163], where tetramer structures based on the *Plasmodium* apical sushi protein were neither toxic nor hemolytic being also highly antigenic and malaria protective when administered in infected rodents, establishing deep differences regarding other previously studied vaccine candidates [164,165].

To start the experimental design, six blood-stage *Plasmodium falciparum* proteins were selected and analyzed by employing a bioinformatics scheme to propose some immunogens representing possible B and T epitopes [166,167,168]. Hence, five native sequences were selected from MSP1, one from MSP2, two from AMA1, three from EBA175, two from SERA, and one from RESA, some of which can be observed in Figure 3A. Once analyzed, key amino-acid pairs in each sequence were selected for introducing a non-natural element into each peptide backbone, in which a specific peptide-bond was replaced with a reduced amide peptide-bond isostere form, as well as L- amino-acids belonging to each pair chosen to be substituted with their D-enantiomers. Thus, about 50 analog sequences were obtained and characterized, as well as employed for animal immunization in order to learn about their immunological functional properties.

This experimental approach consisted of selecting potential antigen sequence targets from *P. falciparum* blood-stage proteins, which were then obtained by solid-phase synthesis and extensively characterized. Introduction of non-natural elements into given amino-acid pairs was then proposed and obtained by liquid and solid-phase synthesis strategies as previously reported [163]. To test the immunological potential of all selected and modified sequences, immunization of groups of four female BALB/c mice of 4 to 6 weeks of age per molecule were conducted using a classical scheme of 0, 15, 30, and 45 days with 50 micrograms of immunogen per animal co-formulated with the Freund’s adjuvant (complete and incomplete versions). Blood samples were collected from each immunized animal before starting the scheme and 8 days after the third and fourth immunizations. ELISA tests lead to establishing which compounds were antigenic by stimulating IgG antibody titers. Therefore, groups of animals that had proven higher antibody titers were split and challenged by i.p administration with 5 × 10^4^ infected RBCs with *P. berghei* ANKA and *P. yoelii* 17XL. A subsequent 20-day follow-up after being challenged lead us to detect those animals which better controlled the parasitemia percentage and even survived the experiment, characterized by the recognition of *P. falciparum* 3D7 and FCB-2 strains lysate proteins by western blot (Figure 4A).

For subsequent experiments, human sera samples from malaria-endemic areas were collected in four different locations of Colombia: 91 from the San Juan Nepomuceno village from the Bolivar province (9°57′0″ N, 75°4′60″ W, and altitude of 156 m above sea level); 72 samples from the Tierralta, village, Cordoba province (8°10′1″ N, 76°4′1″ W and altitude 49 m above sea level); 68 samples from Quibdó city in the Choco province (5°40′59″ N, 76°39′0″ W and altitude 43 m above sea level), and 8 samples from the Tumaco village from the Nariño province (1°48′0″ N, 78°45′0″ W and altitude 3 m above sea level). A control group of 13 samples was collected from a non-endemic malaria area of Colombia, being the Bogotá city (4°37′27″6060 N 74°3′49.1184″ W and 2.630 m above sea level of altitude). For geographical location see Figure 1A. Sampling was allowed by an authorized ethical committee in agreement with international and national regulations. An Informed Consent Statement was obtained from each person who agreed to participate in the study.

Western blot analyses were performed with the whole group of human samples and controls for recognition of *P. falciparum* 3D7 and FCB-2 strains lysate proteins as mentioned above (Figure 4B).

A direct comparison of protein patterns from both mice and human sera reactivity lead us to select those 20 human sera displaying the highest immunochemical profile similarity among both species, and thus subsequent ELISA tests for assaying this selected group of human samples for their capacity of recognizing the immunogens employed for animal vaccination were the chosen step. The sera samples’ control group was also included in this analysis besides a non-modified non-relevant peptide sequence, the so-called CLIP peptide (class II-associated invariant chain peptide).

The results revealed some interesting clues. First, human sera from people from Tierralta, Quibdó, and Tumaco revealed a comparable reactivity by western blot regarding the groups of mice vaccinated with non-natural analogs derived from MSP1, AMA1, and EBA175. Only two samples from San Juan de Nepomuceno displayed some similarity with its rodent counterpart, this fact could be due to this village being the only one in this set that is not located in a high malaria transmission area. Secondly, a remarkable finding is represented by the human sera from Tierralta, Quibdó, and Tumaco (places of high malaria transmission) which show a differential high recognition for most non-natural analogs from selected malaria proteins. Interestingly, the EBA175(E515-H516) analog was the most immunogenic for all groups by displaying the highest OD values as observed (Figure 5A–D), followed by analogs from AMA1(A385-F385), EBA175(D502-R503), and MSP1(K219-L220) and slightly less MSP1(K50-M51) and MSP1(M51-V52). Surprisingly, analogs containing D- amino-acids MSP1(k-1287) and MSP1(p-1288) were highly recognized. Thirdly, none of the native antigen sequences were significantly recognized by the panel of analyzed human sera (data not shown). As observed the MSP1(C1537-P1538) non-natural analog had the lowest reactivity, this belongs to a highly genetically conserved domain of MSP1.

## 4. Conclusions

Difficulties for developing effective vaccines against malaria could arise from our poor understanding of the *Plasmodium* parasite life-cycle, the limited selection criteria for potential targets as immunity inducers, and the multiple immune evasion mechanisms employed by the *Plasmodium* spp. Employing classical approaches for a vaccination with immunogens resembling natural *Plasmodium* antigens presented either as recombinant fused proteins or attenuated forms of parasite stages has proven to be useless for a potent host immune system stimulation. Therefore, specific structure modifications of antigens appear to be relevant for recognition in malaria exposed people, and long-lived antibodies directed especially to EBA175, AMA-1, and N-terminus and central portions of the MSP1 seem to play an important role in humoral immunity to malaria. *Plasmodium* seems to modulate its protein structure at convenience. A lot of work remains to be conducted to analyze the amount of other potential vaccine targets and contribute to overcoming this important public health problem.

## Figures and Tables

**Figure 1 vaccines-09-00115-f001:**
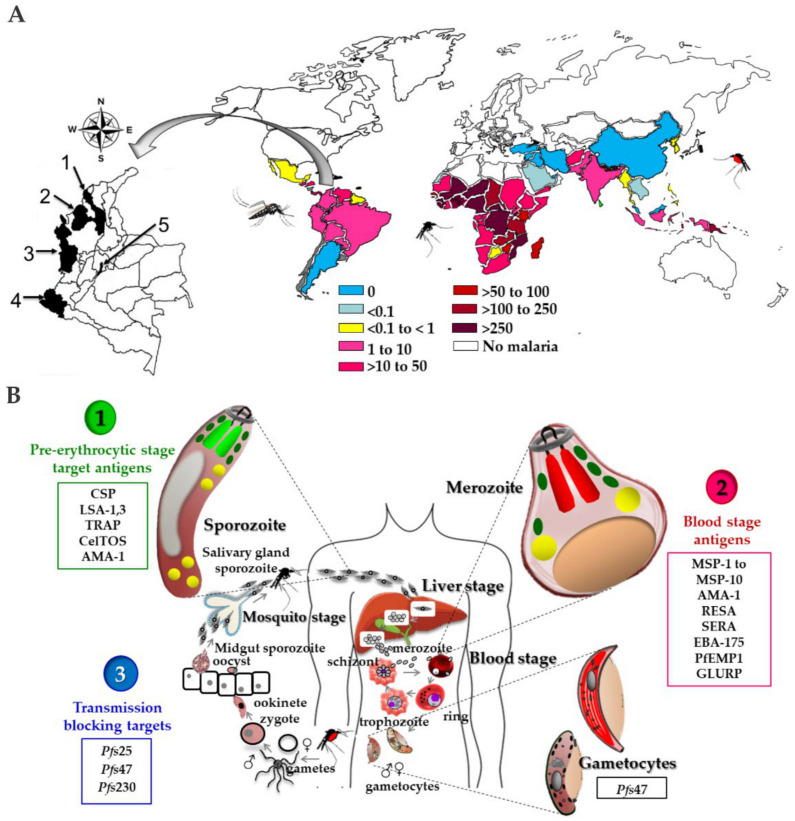
(**A**) Malaria incidence 2019 [1]. Selected villages and provinces of malaria endemic areas in Colombia are 1. San Juan de Nepomuceno (Bolívar), 2. Tierralta (Córdoba),3. Quibdó (Chocó), 4. Tumaco (Nariño) and 5. Bogotá DC (non-endemic for malaria) (**B**) *Plasmodium* spp. life cycle and antigen targets for vaccine development are numbered from 1 to 3.

**Figure 2 vaccines-09-00115-f002:**
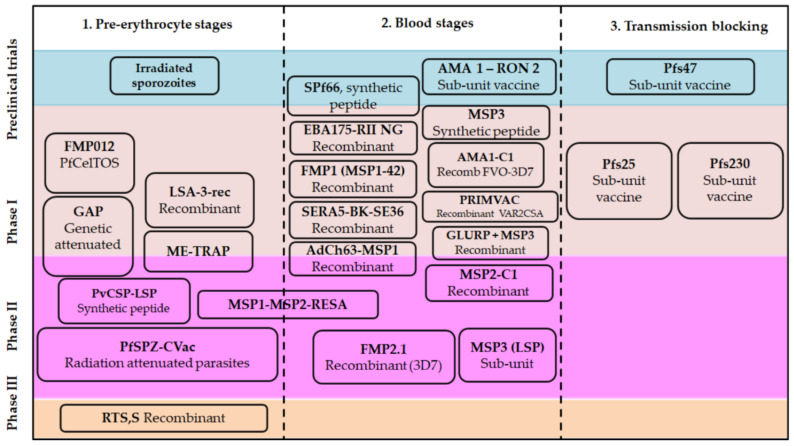
Progress in developing vaccine candidates for malaria.

**Figure 3 vaccines-09-00115-f003:**
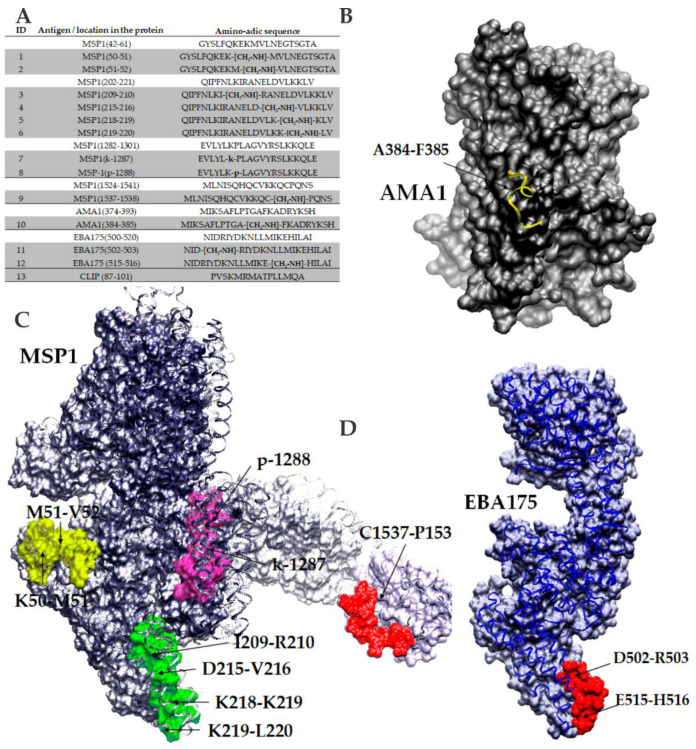
(**A**) *P. falciparum* blood-stages protein targets, amino-acid sequences, and modified analogs. (**B**) Localization of the AMA1 antigen-epitope highlighted in yellow ribbons and its key modified peptide-bond. (**C**) *N*-terminus MSP1 3D structure modelized from a predicted model generated by I-Tasser. *C*-terminal MSP1 fragment pdb file 1ob1 code was employed. Structure recreation shows all peptides sequences. (**D**) Localization of one EBA175 antigen-epitope highlighted in red ribbons and its two key modified peptide-bonds are denoted with arrows. Protein and epitope modeling and personalization were performed with the VMD 1.9.3 version software, from the NIH Biomedical Research Center for Macromolecular Modeling and Bioinformatics, University of Illinois [158]. Protein databank PDB coordinate files used were *4r1c* for AMA-1, *1zro* for EBA175, and *1ob1* for *C*-term MSP1.

**Figure 4 vaccines-09-00115-f004:**
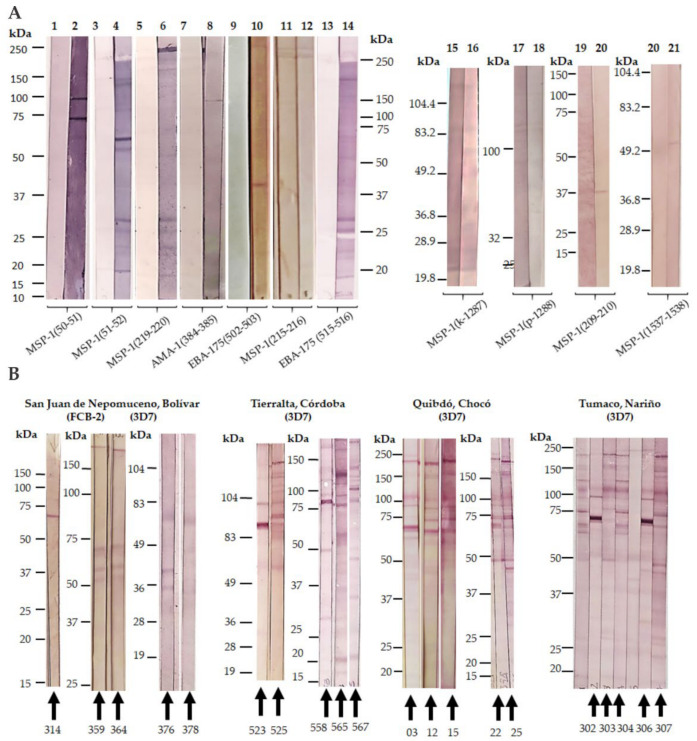
(**A**) Reactivity of serum-antibodies of BALB/c mice immunized with selected blood-stage modified epitopes from *P. falciparum*, versus the 3D7 strain by Western blot. (**B**) Human sera reactivity of samples collected from Colombian malaria endemic areas versus *P. falciparum* 3D7 and FCB-2 strains.

**Figure 5 vaccines-09-00115-f005:**
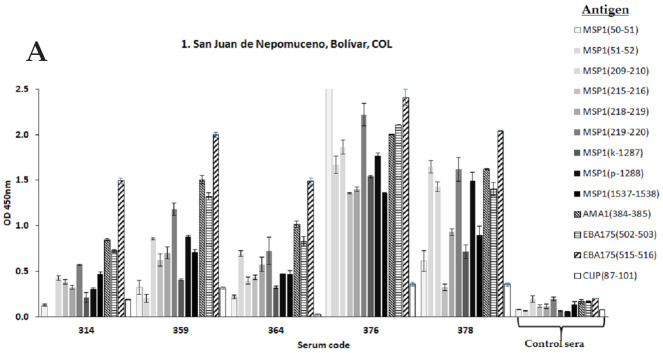
Selected human sera recognizes designed modified-epitope antigens of *Plasmodium* spp. by ELISA tests. (**A**) Samples from San Juan de Nepomuceno village (Bolivar) (**B**) Samples from Tierralta (Cordoba). (**C**) Samples from Quibdó (Chocó) and (**D**) Samples from Tumaco (Nariño). Human sera from a malaria non-endemic areas were obtained from Bogotá DC and used as negative controls. The CLIP endogenous peptide was employed as a non-relevant antigen.

## Data Availability

Not applicable.

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
