# Peer review of "The Search of a Malaria Vaccine: The Time for Modified Immuno-Potentiating Probes"

_vaccines, 2021, doi:10.3390/vaccines9020115_

Round 1

Reviewer 1 Report

This manuscript describes different malaria vaccines under development and tentatively proposes an epitope-based approaches to enhance the vaccine potency. Although this article may be very informative to a large audience, it failed to converge focus on the key issues facing the development of an effective vaccine and the strategies to overcome these challenges. The proposed immune-potentiating approach reported here should be more described, and I don’t think it should be under this format of a review article. The readers may get lost reading this article as I was. Is this review comprehensive (No), critical (No). Comprehensive: the review covers a variety of other topics, that are, in themselves, huge topics; e.g., generalities, life cycle, vectors. Thus, the long section of these topics is not comprehensive and sometimes leaves the reader with the wrong impression of the current state of knowledge. Critical analysis: the review does not provide any critical thinking about why the different malaria vaccine candidates might have obtained different results.

Other comments:

Line 87: Not clear what “2 genetic varieties” is referring to. Remove or provide a sentence explaining the context.

Line 88-89: this sentence refers to the parasite sequestration in the blood vessels. Please correct the sentence for clarity. Also, it is not clear whether “this pathogen” is for P. falciparum or Plasmodium. Need to be specified.

Line 98: “… to survive in various climates ...” Please explain the impact of climates in the Plasmodium vivax parasite survival. I am not aware of this specific notion, so the authors may also provide reference to support this notion. This notion may be more relevant to the mosquito vectors than the parasite itself. Otherwise, this sentence should be modified.

Line 108: Please replace “that can trigger even in death” by “that can lead to death”

Sections 2.1, 2.2 and 2.3 can be combined and summarized in three paragraphs. To my knowledge this is not a book or chapter on Malaria.

Figure 2: RTSS is in a phase III not phase IV trial. Overall, this list of malaria vaccine candidates is far from accurate. There many more vaccine candidates in preclinical stages and two VAR2CSA-based vaccines have been recently tested in phase I trials under the Blood-stage category. Pfs48/45 is also among the transmission blocking candidates. The authors should review this figure with updates candidates or define a criterion by which they include candidates in this figure.

The definition of Cvac as chemically attenuated parasites in Figure 2 is also wrong. Cvac stands for Chemoprophylaxis vaccination and is a vaccination approach with sporozoites. Please correct this in the stable and in the text or provide legend to better explain. Same can be said about GAP. Both are vaccination approaches and not “candidates” and the title of this figure seems to indicate.

Section 3.1: why does the title of this section include “for veterinary …”. The authors should be focus on the subject they plan to discuss in this review and not be all over the place. This section can also be summarized.

Sub-section 3.2.2: this section should be amended with VAR2CSA antigen

Sub-section 3.2.3: It looks to me that this sub-title should be remove as these antigens are also “blood stage antigen” and therefore should be listed in the section 3.2.2.

Lines 717-791: Are the data presented in these paragraphs previously published? If yes, please provide the reference summarize the findings instead.

Author Response

Reviewer #1

Comments and Suggestions for Authors

This manuscript describes different malaria vaccines under development and tentatively proposes an epitope-based approaches to enhance the vaccine potency. Although this article may be very informative to a large audience, it failed to converge focus on the key issues facing the development of an effective vaccine and the strategies to overcome these challenges. The proposed immune-potentiating approach reported here should be more described, and I don’t think it should be under this format of a review article. The readers may get lost reading this article as I was. Is this review comprehensive (No), critical (No). Comprehensive: the review covers a variety of other topics, that are, in themselves, huge topics; e.g., generalities, life cycle, vectors. Thus, the long section of these topics is not comprehensive and sometimes leaves the reader with the wrong impression of the current state of knowledge. Critical analysis: the review does not provide any critical thinking about why the different malaria vaccine candidates might have obtained different results.

Answer/ Thank you for your valuable and highly relevant comment which certainly has set an important challenge to us for improving our Review. In consequence, we took this constructive criticism as a starting point to lead us to obtain a carefully adjusted and revised version of this work. As you will see in the revised version thereof strong modifications to the text, figures and tables were performed to allow a more interesting version of this review in which our main findings regarding novel and potential malaria vaccine candidates are highlighted.

Other comments:

Comment #1:

Line 87: Not clear what “2 genetic varieties” is referring to. Remove or provide a sentence explaining the context.

Answer/ In agreement and considering that this information is not relevant for the manuscript, we have decided to remove it from the text.

Comment #2:

Line 88-89: this sentence refers to the parasite sequestration in the blood vessels. Please correct the sentence for clarity. Also, it is not clear whether “this pathogen” is for P. falciparum or Plasmodium. Need to be specified.

Answer/ In agreement, the phrase was reformulated for more clarity and understanding.

Comment #3:

Line 98: “… to survive in various climates ...” Please explain the impact of climates in the Plasmodium vivax parasite survival. I am not aware of this specific notion, so the authors may also provide reference to support this notion. This notion may be more relevant to the mosquito vectors than the parasite itself. Otherwise, this sentence should be modified.

Answer/ Literature for this item can be found in references 14 and 15 in the References section. To consider a better understanding and avoiding any confusion, this statement was adjusted in the revised version of the manuscript.

Comment #4:

Line 108: Please replace “that can trigger even in death” by “that can lead to death”

Answer/ In line with this comment, this statement was adjusted.

Comment #5:

Sections 2.1, 2.2 and 2.3 can be combined and summarized in three paragraphs. To my knowledge this is not a book or chapter on Malaria.

Answer/ Given the relevance of this subject, we consider preserving the integrity of the mentioned subsections of the adjusted version of this Review.

Comment #6:

Figure 2: RTSS is in a phase III not phase IV trial. Overall, this list of malaria vaccine candidates is far from accurate. There many more vaccine candidates in preclinical stages and two VAR2CSA-based vaccines have been recently tested in phase I trials under the Blood-stage category. Pfs48/45 is also among the transmission blocking candidates. The authors should review this figure with updates candidates or define a criterion by which they include candidates in this figure.

Answer/ Thank you for this comment. After verifying, Figure 2 was enriched regarding an update for its content in agreement with suggestions and supplementary material attached to the submission.

Comment #7:

The definition of Cvac as chemically attenuated parasites in Figure 2 is also wrong. Cvac stands for Chemoprophylaxis vaccination and is a vaccination approach with sporozoites. Please correct this in the stable and in the text or provide legend to better explain. Same can be said about GAP. Both are vaccination approaches and not “candidates” and the title of this figure seems to indicate.

Answer/ In line with this comment Figure, 2 was updated and complemented accordingly with supplementary material information provided in this Review. Thus, a new version of Figure 2 has been modified including more examples and information, a multistage vaccine candidate, and those candidates that completed Phase I and introduced to Phase II were included as can be seen in the figure

Comment #8:

Section 3.1: why does the title of this section include “for veterinary …”. The authors should be focus on the subject they plan to discuss in this review and not be all over the place. This section can also be summarized.

Answer/ Thank you for this comment. We agree that this title is wrongly expressed on the main discussion subject. Thus, it was modified in the revised version of the manuscript to focus only on human permitted adjuvants. Therefore, the title of this section was adjusted and now presented as Adjuvants and Vaccine Formulations.

Comment #9:

Sub-section 3.2.2: this section should be amended with VAR2CSA antigen.

Answer/ As it was kindly proposed, information regarding VAR2CCSA antigen and PAMVAC and PRIMVAC vaccine candidates were included in this section and a new reference also was included (Ref#133). Figures 2 and 3 were modified accordingly.

Comment #10:

Sub-section 3.2.3: It looks to me that this sub-title should be remove as these antigens are also “blood stage antigen” and therefore should be listed in the section 3.2.2.

Answer/ Thank you for this comment. We have decided to leave this subsection as initially proposed since Plasmodium spp organelle-derived antigens are being classically regarded as a specific matter.

Comment #11:

Lines 717-791: Are the data presented in these paragraphs previously published? If yes, please provide the reference summarize the findings instead.

Answer/ We are presenting by the very first time the results that are shown from pages 717 to 791 of the manuscript. Cited references in this section are aimed to orientate the Vaccines MDPI readers about experimental design and previously reported findings, as well as the conceptual basis of our research.

Reviewer 2 Report

José Manuel Lozano group has comprehended “The Search of a Malaria Vaccine: The Time for Modified Immuno-Potentiating Probes”. This is a very exhaustive manuscript. The explanation and literature on Plasmodium spp life cycle is very well written and covered, respectively.  

However, few parts need to be improved for better understanding.

  • In many instances sentence is overcrowded, e.g., lines 43-47; lines 48-51, and so on. It is better to provide short sentences.
  • Overall the references are not fully covered. These are some pertinent references should be cited.

Immunity response in malaria section

https://doi.org/10.1038/s41577-019-0158-z

https://doi.org/10.1016/j.coi.2016.06.008

https://doi.org/10.1002/eji.1830200706

https://doi.org/10.4049/jimmunol.172.4.2487

https://doi.org/10.1073/pnas.88.18.7963

Adjuvants section

https://doi.org/10.1038/nrmicro1681

https://doi.org/10.1016/j.tips.2017.06.002

https://doi.org/10.1016/j.intimp.2019.105684

https://doi.org/10.1111/imr.12889

https://doi.org/10.1016/j.coi.2011.03.009

https://doi.org/10.1016/j.smim.2018.05.001

  • Figures are blurred in printed pages; please take care of the resolution. The font size used in Figure 1b is not et al. visible.
  • Line 326: “third, the acquired immunity specifically in the populations from endemic areas” is this comes under the immuo-prophylactic treatment? Please have a look. It is a natural process, not a treatment strategy.
  • Line 373: “number of contacts” ------ “number of doses’
  • Line 381: Mechanism of action of “delivery systems” has been poorly defined. Please rewrite it.
  • Line 625: names should be in italics.

Author Response

Reviewer #2

Comments and Suggestions for Authors

José Manuel Lozano group has comprehended “The Search of a Malaria Vaccine: The Time for Modified Immuno-Potentiating Probes”. This is a very exhaustive manuscript. The explanation and literature on Plasmodium spp life cycle is very well written and covered, respectively.  

Answer/ We appreciate very much your positive and encouraging comment.

However, few parts need to be improved for better understanding.

  • Comment #1. In many instances sentence is overcrowded, e.g., lines 43-47; lines 48-51, and so on. It is better to provide short sentences.

Answer/ In line with this comment, a deep language grammar revision was performed on the whole manuscript to lead us to propose an adjusted version thereof.

  • Comment #2. Overall the references are not fully covered. These are some pertinent references should be cited.

Immunity response in malaria section

https://doi.org/10.1038/s41577-019-0158-z

https://doi.org/10.1016/j.coi.2016.06.008

https://doi.org/10.1002/eji.1830200706

https://doi.org/10.4049/jimmunol.172.4.2487

https://doi.org/10.1073/pnas.88.18.7963

Adjuvants section

https://doi.org/10.1038/nrmicro1681

https://doi.org/10.1016/j.tips.2017.06.002

https://doi.org/10.1016/j.intimp.2019.105684

https://doi.org/10.1111/imr.12889

https://doi.org/10.1016/j.coi.2011.03.009

https://doi.org/10.1016/j.smim.2018.05.001

Answer/ All suggested references have been included in the revised version of the manuscript.

  • Comment #3. Figures are blurred in printed pages; please take care of the resolution. The font size used in Figure 1b is not et al. visible.
  • Answer/ In agreement with this comment, we have worked out every presented Figure in this Review to improve their content and graphical quality. Figure 1b has been adjusted accordingly.
  •  
  • Comment #4. Line 326: “third, the acquired immunity specifically in the populations from endemic areas” is this comes under the immuo-prophylactic treatment? Please have a look. It is a natural process, not a treatment strategy.
  • Answer/ After carefully revised, this sentence was complemented to highlight the relevance of vaccination.
  •  
  • Comment #5. Line 373: “number of contacts” ------ “number of doses’
  • Answer/ As kindly suggested by the reviewer, this sentence was adjusted for a better understanding and clarify some differences between dose and number of contacts concepts in immunity.

  • Comment #6. Line 381: Mechanism of action of “delivery systems” has been poorly defined. Please rewrite it.
  • Answer/ As recommended, this statement has been re-written, and the delivery system concept was properly defined for a better understanding.
  •  
  • Comment #7. Line 625: names should be in italics.
  • Answer/ Corrections were made accordingly.

Round 2

Reviewer 1 Report

In Fig 2, the authors indicated PAMVAC and PRIMVAC but in the text they only referred to PRIMVAC. Lines 614-617), the claim that PRIMVAC was the first of such vaccine in trial is technically not correct. First in human report of var2csa based vaccine was described with PAMVAC. Please provide the corresponding references in Fig 2 and the text.

I still think the list of candidates in preclinical studies is incomplete particularly in TBV.

Author Response

We appreciate very much your valuable comments and suggestions which certainly have helped to improve the present work´s quality

Reviewer #1

Comments and Suggestions for Authors

In Fig 2, the authors indicated PAMVAC and PRIMVAC but in the text they only referred to PRIMVAC. Lines 614-617), the claim that PRIMVAC was the first of such vaccine in trial is technically not correct. First in human report of var2csa based vaccine was described with PAMVAC. Please provide the corresponding references in Fig 2 and the text.

I still think the list of candidates in preclinical studies is incomplete particularly in TBV.

Answer/ Thank you for your valuable suggestions and comments. In agreement, the text was adjusted (lines 614 to 617) to PRIMVAC vaccine candidate as the reference argument based on a recently published work (reference number 133 by Sirima, S.; et al., Lancet Infect Dis, 2020, 20(5), pp. 585-597, doi: https://doi.org/10.1016/S1473-3099(19)30739-X. As suggested Figure 2 was adjusted by including a reference number.